# Weakly Supervised Deep Functional Map for Shape Matching

**Abhishek Sharma**
LIX, École Polytechnique
`kein.iitian@gmail.com`

**Maks Ovsjanikov**
LIX, École Polytechnique
`maks@lix.polytechnique.fr`

## Abstract

A variety of deep functional maps have been proposed recently, from fully supervised to totally unsupervised, with a range of loss functions as well as different regularization terms. However, it is still not clear what are minimum ingredients of a deep functional map pipeline and whether such ingredients unify or generalize all recent work on deep functional maps. We show empirically the minimum components for obtaining state-of-the-art results with different loss functions, supervised as well as unsupervised. Furthermore, we propose a novel framework designed for both full-to-full as well as partial to full shape matching that achieves state of the art results on several benchmark datasets outperforming, even the fully supervised methods. Our code is publicly available at `https://github.com/Not-IITian/Weakly-supervised-Functional-map`

## 1 Introduction

Shape correspondence is a fundamental problem in computer vision, computer graphics and related fields since it facilitates many applications such as texture or deformation transfer and statistical shape analysis Bogo et al. [2014] to name a few. While classical correspondence methods have been based on handcrafted features or deformation models Van Kaick et al. [2011], more recent approaches have focused on *learning* an optimal model directly from 3D data. This includes approaches based on template fitting and reconstruction Groueix et al. [2018, 2019], and methods that exploit different definitions of convolution and phrase correspondence as a dense labeling problem Wei et al. [2016], Masci et al. [2015], Boscaini et al. [2016] among others.

A prominent direction in learning-based shape matching was pioneered by the FMNet work, Litany et al. [2017a] by exploiting the *functional map representation* Ovsjanikov et al. [2012] and learning features that recover optimal functional maps rather than e.g. individual point labels. The use of the functional map representation allows to efficiently impose global correspondence constraints, and has been recently been extended in both unsupervised Halimi et al. [2019], Roufosse et al. [2019] and supervised settings Donati et al. [2020]. Despite significant progress in this area, there still exist three major issues. First, the most accurate recent approach Donati et al. [2020] is limited to supervised setting that requires ground truth correspondences that are difficult to obtain considering the cost of annotating a dense point-to-point map on each shape pair. Second, despite a variety of deep functional maps-based methods, it is still not clear what are minimum ingredients of a deep functional map pipeline. More importantly, do such minimum ingredients unify or generalize all recent work on deep functional maps. While a battery of loss functions and regularization have been proposed for different deep functional maps, as we demonstrate below, the devil is not in the loss functions. Instead, using a low number of Laplacian eigen-basis, very weak supervision in the form of rigid alignment and enforcing basic structural properties of resulting functional map are sufficient to obtain high quality results. Moreover, our approach generalize to many loss functions proposed recently and does not require Geodesic matrices, as in FMnet Litany et al. [2017a] and UnSupFmnet Halimi et al. [2019],

ground truth maps, as in GeomFmap Donati et al. [2020] and FMnet, regularizers, such as descriptor preservation in SurfmNet Roufosse et al. [2019] and regularized FMap layer in GeomFMap Donati et al. [2020]. Third, recent learning-based approaches are neither designed nor tested for the *partial shape matching problem* [Rodolà et al., 2017, Litany et al., 2017b] which is of great interest in robotics [Chavdar et al., 2012] and Virtual reality applications [Sharma et al., 2016]. To this end, we propose a weakly supervised framework that addresses all three major issues.

In this paper, weak supervision implies that datasets are only approximately rigidly aligned, which is necessary primarily due to the presence of symmetries. Since some poses (e.g. the neutral pose) are fully extrinsically symmetric, a PointNet like feature extractor cannot distinguish left/right unless the shapes are aligned. Interestingly, as we demonstrate below, such weak supervision is sufficient to obtain high quality results.

Broadly, there are three main components to any deep functional map pipeline, namely feature extractor, choice of basis functions and design of empirical loss or regularization on the functional map. In this paper, we make contributions on all three fronts. First, we propose to learn feature descriptors directly from raw data with a very weak supervision and establish that for non-rigid shape correspondence, rigid alignment supervision turns out to be sufficient to obtain accurate results. Remarkably, it also outperforms the fully supervised state-of-the-art methods, which rely on ground truth point-to-point correspondences, on challenging benchmarks. Secondly, we show that the combination of our feature extractor projected to low number of Laplacian eigen basis (30) and unsupervised loss, consisting of simple regularization terms, suffice to obtain state of the art result for any recently proposed loss functions. Thirdly, to address partial shape matching, we propose a novel data driven method to learn an optimal alignment between source and target Laplacian eigen basis functions which paves the way for future work on deep functional maps in partial shape matching.

## 2   Related Work

**Functional Maps**   Computing point-to-point maps between two 3D discrete surfaces is a very well-studied problem. We refer to a recent survey Sahillioğlu [2019] for an in-depth discussion. Our method builds upon the functional map pipeline, introduced in Ovsjanikov et al. [2012] and then significantly extended in follow-up works (see, e.g., Ovsjanikov et al. [2017]). Functional maps encode correspondences as small matrices, expressed in a reduced basis, which greatly simplifies the associated optimization problems. A range of recent works, including Kovnatsky et al. [2013], Huang et al. [2014], Burghard et al. [2017], Rodolà et al. [2017], Nogneng and Ovsjanikov [2017], Ren et al. [2018] among many others, have extended the generality and improved the robustness of the functional map estimation pipeline, by suggesting regularizers, robust penalties and powerful post-processing approaches. Nevertheless, existing non-learning based methods are strongly tied to the choice of descriptor (also known as "probe") functions, which must be specified manually a priori. We also note that there also exist other techniques that learn correspondences without using the functional map representation, e.g., Wei et al. [2016], Boscaini et al. [2016], Monti et al. [2017]. However, such techniques typically either require significantly more training data (essentially because they treat shape correspondence as a dense labeling problem with a very large number of labels), or do not learn from 3D geometry which is the main goal of this paper.

**Supervised Learning from raw 3D shape**   In contrast to axiomatic approaches that use hand-crafted features, a variety of methods have also been proposed to *learn* the optimal features or descriptors from 3D data. In the functional maps domain, this was first suggested in Corman et al. [2014] using classical optimisation techniques and then in the seminal Deep Functional Maps work Litany et al. [2017a] that proposed a deep learning architecture called FMNet to compute optimal features from data. This architecture was based on optimizing a non-linear transformation of SHOT descriptors, Tombari et al. [2010] to obtain maps that are as close as possible to given ground truth correspondences. Follow-up works have extended this approach to the unsupervised setting Roufosse et al. [2019], Halimi et al. [2019] by modifying the training loss, but still used pre-defined descriptors for optimization. These methods generalize poorly across datasets as the input features such as SHOT descriptors are sensitive to the triangle mesh structure, which can vary drastically across different datasets.

Most recently, works including Groueix et al. [2018], Donati et al. [2020] have shown that feature functions can be learned directly from the *raw 3D data* without relying on pre-defined descriptors,

resulting in a significantly more robust and accurate methods. However, to obtain good results these works had to rely on ground truth correspondences and do not generalize its empirical success beyond its own setup. Although PointNet Qi et al. [2017a] and its variants (Qi et al. [2017b]) achieve impressive results from raw point clouds for classification tasks, they are not yet competitive for shape correspondence task.

**Partial Shape Matching**    While some formulations of functional maps allow to deal with the lack of isometry and partiality, this framework is in principle not designed to deal with partial correspondence. Rodolà et al. [2017] provided an empirical evidence and theoretical analysis of a surprising property of interaction between Laplacian eigenfunctions as a result of removing parts from surfaces. This implies that there exists an unknown alignment between eigenfunctions of partial shapes and full shapes and knowing it results in a special slanted diagonal structure of the correspondence matrix. However, their solution relies on a complicated alternating optimization over the spectral domain and the spatial domain. Instead, Litany et al. [2017b] proposed an efficient and fully spectral domain method for finding this transformation matrix between the two eigen spaces. This approach, however, is still based on hand-crafted features, optimization on Stiefel manifold and is instance specific. Besides, replacing handcrafted features by learnable feature descriptors is not straightforward due to manifold optimization involved in the process. We address both these issue by proposing a novel method that mitigates these issues by learning directly from raw data.

## 3   Background

As mentioned above, in this work we focus on the functional map representation, due to its efficiency, and rich geometric structure. Before describing our method we provide a brief overview of the basic pipeline.

**Basic Deep Functional Map Pipeline**    Given a source and a target shape, $S_1, S_2$, containing, respectively, $n_1$ and $n_2$ vertices, the basic pipeline for computing a map between them using the functional map framework and its deep counterparts is as follows, Ovsjanikov et al. [2017]:

1. On each shape, compute a small set of $k_1, k_2$ of basis functions , e.g. by taking the first few eigenfunctions of the respective Laplace-Beltrami operators.
2. Compute a set of descriptor (also known as "probe") *functions* on each shape that should be preserved by the unknown map. In our case, these functions are PointNet features that are further projected onto the first 30 Laplacian eigen functions and their coefficients are stored as columns of matrices $\mathbf{A}, \mathbf{B}$.
3. Compute the optimal *functional map* $\mathbf{C}$ by solving the following optimization problem:

$$C_{\text{opt}} = \arg\min_{\mathbf{C}} E_{\text{desc}}(\mathbf{C}) + \alpha E_{\text{reg}}(\mathbf{C}), \tag{1}$$

   where $E_{\text{desc}}(\mathbf{C}) = \left\| \mathbf{CA} - \mathbf{B} \right\|^2$ aims at the descriptor preservation whereas the second term acts as a regularizer on the map by enforcing its overall structural properties. Eq (1) can be solved with any convex solver. However, when descriptor functions are neural network based, we need to differentiate the solution with respect to the spectral features $\mathbf{A}, \mathbf{B}$, which is challenging when $\mathbf{C}$ is computed via an iterative solver. Deep functional maps avoid this by only optimizing $\mathbf{C}$ the first part of the energy : $\left\| \mathbf{CA} - \mathbf{B} \right\|^2$ by solving a simple linear system for which the derivatives can be computed in closed form.
4. The functional map $\mathbf{C}$ in spectral domain is then converted to a point-to-point spatial map using one of several techniques e.g nearest neighbor search in the spectral embedding.

## 4   Method

### 4.1   Overview

In this section, we first introduce our approach to learning descriptors from raw 3D shapes for full to full shape matching. Afterwards, we detail our novel partial shape matching algorithm that learns an optimal alignment of Laplacian eigen basis functions, given the spectrum of partial and full shape. Note that the feature descriptor extraction is common to both approaches. However, our unsupervised loss function is totally different for partial and full shape matching.

## 4.2 Weak Supervision

In both full and partial matching cases, our method is "weakly supervised" in the sense that we expect the input non-rigid shapes to be approximately *rigidly* aligned. This means having a consistent 'up' direction (along, e.g., the $y$ axis) and an approximate forward-facing direction (along, e.g., the $z$ direction). Some existing datasets, such as partial SHREC, Cosmo et al. [2016], already satisfy this assumption. When considering multiple datasets, we only need to make sure that these axes are consistent, which can be done with very little manual intervention. We stress that we *do not use* ground truth point-to-point or functional correspondences, and that obtaining reliable detailed ground truth maps requires significant effort especially when considering cross-dataset learning. Weak supervision is necessary primarily due to the presence of symmetries. Since some poses (e.g. the neutral pose) are fully extrinsically symmetric, *some way* to disambiguate left/right is necessary for accurate correspondence. In our case, we exploit weak supervision in the form of rigid alignment, and further use it explain the performance drop of a fully supervised methods like GeomFmap, Donati et al. [2020], when trained on aligned dataset and tested on Scape (non-aligned).

## 4.3 Feature extractor

Our main goal is to learn functional characterizations of point clouds that will later be used to compute spectral descriptors and then functional maps. Thus, this network must be applied with the same weights to the source and target shapes in a Siamese way using shared learnable parameters. Our feature extractor is based on Pointnet $++$, Qi et al. [2017b] that extracts local features capturing fine geometric structures from small neighborhoods. Such local features are further grouped into larger units and processed to produce higher level features. Our feature extraction network is based on the standard architecture consisting of $4$ sampling layers, with first layer sampling $1024$ points and $4$ feature propagation layers such that final layer outputs $128$ dimension feature descriptor for each input shape. For details on all the parameters of network, please see the source code.

## 4.4 Unsupervised loss for full shape matching

Given the extracted feature functions, we first project them onto the Laplacian basis and then compute the optimal functional map by minimizing $\min_{\mathbf{C}} \left\| \mathbf{CA} - \mathbf{B} \right\|^2$. As noted in Litany et al. [2017a], this leads to a simple linear system of equation, whose solution can be differentiated during training. We therefore train our feature extraction network by imposing an unsupervised loss on the optimized functional map. Our loss follows the approach of Roufosse et al. [2019] and is based on three key structural properties of a functional map between two approximately isometric shapes.

**Bijectivity** Transporting functions on a shape and transporting them back should yield the same functions. Following Eynard et al. [2016], Roufosse et al. [2019], we therefore enforce that composition between $\mathbf{C_{12}}$ and $\mathbf{C_{21}}$ to be as closely as possible to $\mathbf{I}$, the identity matrix, which leads to: $E_1 = \|\mathbf{C}_{12}\mathbf{C}_{21} - \mathbf{I}\|^2 + \|\mathbf{C}_{21}\mathbf{C}_{12} - \mathbf{I}\|^2$

**Orthogonality** As observed in the functional map literature, Ovsjanikov et al. [2012], Rustamov et al. [2013], Roufosse et al. [2019] a point-to-point map is locally area preserving if and only if the corresponding functional map is *orthonormal*. Thus, for shape pairs, approximately satisfying this assumption, a natural penalty in our unsupervised pipeline is: $E_2 = \|\mathbf{C}_{12}^{\top}\mathbf{C}_{12} - \mathbf{I}\|^2 + \|\mathbf{C}_{21}^{\top}\mathbf{C}_{21} - \mathbf{I}\|^2$

**Laplacian commutativity** Having functional maps that commute with the Laplace-Beltrami operators is known to be a common regularizer in the functional map pipeline Rosenberg [1997], Ovsjanikov et al. [2012]. We recall that this constraint helps find better mappings since it promotes near-isometric point-to-point maps: $E_3 = \left\|\mathbf{C}_{12}\mathbf{\Lambda}_1 - \mathbf{\Lambda}_2\mathbf{C}_{12}\right\|^2 + \left\|\mathbf{C}_{21}\mathbf{\Lambda}_2 - \mathbf{\Lambda}_1\mathbf{C}_{21}\right\|^2$ where $\mathbf{\Lambda}_1$ and $\mathbf{\Lambda}_2$ are diagonal matrices of the Laplace-Beltrami eigenvalues on the two shapes.

Thus, our unsupervised loss function is a combination of all three structural properties and weighted as follows: $L = E_1 + E_2 + 0.001E_3$ where the weighing scalars are found empirically.

## 4.5 Basis Alignment for Partial Shape Matching

The basic pipeline described above for shape matching breaks down in the case of partial shape matching. This is primarily because structural properties of map such as bijectivity, area preservation

(orthogonality) are not applicable anymore. Rodolà et al. [2017] showed that for each "partial" eigenfunction (i.e., each eigenfunction of a partial shape), there exists a corresponding "full" eigenfunction (i.e., some eigenfunction of the full shape). The problem then reduces to finding alignment in $k$ dimensional eigen space that is achieved by optimizing for a new basis on one shape only and keeping the other fixed to the standard Laplacian eigenfunctions. Due to this coupling, the new basis functions will behave consistently resulting in almost perfectly diagonal C even in the absence of a perfect isometry. Keeping the same notation as before, where $\mathbf{A}$ and $\mathbf{B}$ represents the pointnet++ descriptors projected onto the laplacian eigen basis, it is written as follows:

$$\min_{\mathbf{X}} \left\| \mathbf{A_r} - \mathbf{X}^\top \mathbf{B} \right\|^2 + \text{off}(\mathbf{X}^\top \mathbf{\Lambda X}), \tag{2}$$

where $\mathbf{A_r}$ contains the $r \times k$ submatrix of $\mathbf{A}$ (the first $r$ rows of matrix $\mathbf{A}$) and $\mathbf{X}$ of size $k \times r$ is a transformation matrix between the two eigen spaces that stores the coefficients of desired linear combination. $\mathbf{\Lambda}$ is a diagonal matrix of the first $k$ eigenvalues of partial shape. The second term in Eq. (2) is a regularizer on $\mathbf{X}$ that ensures that resulting eigen basis functions on partial shape minimize the Dirchelet energy on its Laplacian Beltrami operator $\Delta$. The value of $r$ is estimated from the spectrum of partial and full shape as follows: $r = \max_{i=1}^{k_p} \{i \mid \lambda_i^p < \max_{j=1}^{k_f} \lambda_j^f \}$ after setting $k_p = k_f = 60$ where f denotes the full shape and p denotes the partial one. We upper bound the rank obtained by $40$.

The method of Litany et al. [2017b] obtains the descriptor function matrix $\mathbf{A}$ and $\mathbf{B}$ using precomputed SHOT descriptors. Besides, it constrains $\mathbf{X}$ to be an orthogonal matrix and thus optimize it using manifold optimization solver on Stiefel Manifold. However, we do not impose any orthogonality constraint on $\mathbf{X}$ and optimize Eq. (2) differently since our descriptor functions are PointNet $++$ based and need to be learned simultaneously. So, instead of optimzing over $\mathbf{X}$, we are optimizing the functional over $\mathbf{X}$, $\mathbf{A}$ and $\mathbf{B}$.

$$\min_{\mathbf{X,A,B}} \left\| \mathbf{A_r} - \mathbf{X}^\top \mathbf{B} \right\|^2 + \text{off}(\mathbf{X}^\top \mathbf{\Lambda X}), \tag{3}$$

We split the functional in Eq. 3 in two parts and first optimize for $\mathbf{X}$ by solving $\left\| \mathbf{A_r} - \mathbf{X}^\top \mathbf{B} \right\|^2$ with a simple linear system for which the derivatives can be computed in closed form. Given this optimal $\mathbf{X}$, we then impose the loss on $\mathbf{X}$ by computing the second part of Eq (3) and use this unsupervised loss to backpropagate gradients to learn the appropriate descriptor functions. Note that for partial matching this loss term is *the only* one we use, whereas in the full shape matching setting we use a more powerful loss described in Section 4.4.

**Implementation** We implemented our method in TensorFlow Abadi et al. [2015]. We train our network with a batch size of $24$ shape pairs for $10000$ steps. We use a learning rate of $1e-4$ with Adam optimizer. During training, we randomly sample $4000$ points from each shape while training with Surreal dataset whose shapes contain $7000$ points each. For other datasets such as Scape and Faust remesh, that contain roughly $5000$ points each, we randomly sample $3000$ points during for training. Since partial shape dataset contains very limited number of shapes, we describe its experimental setup later in Section 5.3. For a fair comparison with some baseline methods, we use a very recent and efficient refining algorithm, called ZoomOut Melzi et al. [2019] based on navigating between spatial and spectral domains while progressively increasing the number of spectral basis functions.

## 5 Results

This section is divided into three subsections where each provides a separate evaluation of our contributions. Section 5.1 shows the experimental comparison of our weakly supervised approach with fully supervised state-of-the art methods for near-isometric shape matching. Section 5.2 demonstrates that weak rigid alignment of datasets, low number of Laplacian eigen basis and enforcing structural properties of a map suffice to obtain excellent results across a variety of loss functions. Finally, Section 5.3 demonstrates the effectiveness of our novel partial shape matching framework. We evaluate all results by reporting the per-point-average geodesic distance between the ground truth map and the computed map. All results are multiplied by 100 for the sake of readability.

Table 1: Results on remeshed Faust and Scape.

| Method \ Dataset | F | S | F on S | S on F |
|---|---|---|---|---|
| SURFMNet | 15. | 12. | 32. | 32. |
| SURFMNet+icp | 7.4 | 6.1 | 19. | 23. |
| Unsup FMNet | 10. | 16. | 29. | 22. |
| Unsup FMNet+pmf | 5.7 | 10. | 12. | 9.3 |
| FMNet | 11. | 17. | 30. | 33. |
| FMNet+pmf | 5.9 | 6.3 | 11. | 14. |
| 3D-CODED | 2.5 | 31. | 31. | 33. |
| GeomFmap | 3.1 | 4.4 | 11. | 6.0 |
| GeomFmap +zo | **1.9** | **3.0** | 9.2 | **4.3** |
| Ours | 3.3 | 7.3 | 11.7 | 6.2 |
| Ours + zo | **1.9** | 4.9 | **8.0** | **4.3** |

Table 2: Results when trained on Surreal and tested on remeshed Faust and Scape.

| Method \ Dataset | F | S |
|---|---|---|
| GeomFmap +Zo | **2.5** | 9.2 |
| 3D-CODED | 4.9 | 6.0 |
| Ours | 5.0 | 8.3 |
| Ours+zo | **2.8** | **5.5** |

## 5.1 Near-isometric Shape Matching

In this section we evaluate our method for complete (full to full) near isometric shape matching. We compare our method with state-of-the-art approaches while focusing especially on the the very recent functional map-based technique Donati et al. [2020], which was shown to outperform existing competitors.

**Datasets** For a fair comparison with Donati et al. [2020], we follow the same experimental setup and test our method on a wide spectrum of datasets: first, the re-meshed versions of FAUST dataset Bogo et al. [2014] and the SCAPE Anguelov et al. [2005], made publicly available by Ren et al. [2018]. Lastly, we also use the training dataset of 3D-CODED, consisting in 230K synthetic shapes generated using SURREAL Varol et al. [2017] with the parametric model SMPL introduced in Loper et al. [2015]. We use a subset of it for training purposes to compare the generalization ability of different methods to changes in connectivity and triangulation. This is achieved by training on this synthetic data and testing on re-meshed datasets such as FAUST and SCAPE.

**Baselines** We compare our method to several state of the art methods: the first category includes a variety of unsupervised deep functional maps proposed recently with SHOT descriptors. Second category includes supervised methods that directly learn from 3D data. This includes the supervised template based approach of 3D-CODED Groueix et al. [2018] as well as the recent work GeomFmap Donati et al. [2020]. All baseline results are taken from Donati et al. [2020]. In the case of SHOT based deep functional maps Litany et al. [2017a], Halimi et al. [2019], Roufosse et al. [2019], all results are invariant by any rigid transformation of the input shapes and therefore, no alignment is required. For a fair comparison with other methods, we show our results with and without ZoomOut Melzi et al. [2019] refinement, referred to as ZO. For conciseness, we refer to our method as Ours in the following text. We compare these different methods in Table 1.

Table 3: Ablation study of individual losses with and without alignment when trained with Surreal.

| Losses | All | E1 | E2 | E3 | (E1+E3) | All-not-aligned |
|---|---|---|---|---|---|---|
| **Scape** | **8.3** | 13 | 16 | 10.5 | 9.2 | 22 |
| **Faust** | **5.0** | 11 | 14 | 9.0 | 6.3 | 8.0 |

**Generalization Experiments** Following the standard protocol, we split FAUST re-meshed and SCAPE re-meshed into training and test sets containing 80 and 20 shapes for FAUST, and 51 and 20 shapes for SCAPE. **F** and **S** in Table 1 shows the results for training and testing on same dataset, FAUST and SCAPE, respectively whereas **F on S** means trained on FAUST and tested on SCAPE. In Table 2, results are shown with the SURREAL dataset from which we sample 500 shapes for training and test the trained models on test sets of FAUST re-meshed, SCAPE re-meshed. We compare with 3D-CODED and GeomFMap since they outperform every other method and learn from raw 3D geometry. We report baseline numbers from Donati et al. [2020] which report performance of different methods by varying the size of training set from few hundred to thousands. We pick the best results obtained with any number of shapes. All results are multiplied by 100 for the sake of readability. We also report results without ZoomOut refinement.

**Results and Discussion** As evident in Table 1, our method performs on par with the fully supervised approaches such as 3D-CODED Groueix et al. [2018] and GeomFMap Donati et al. [2020]. We observe comparable or superior performance to the supervised approach in Table 2. We obtain a particularly remarkable performance on the SCAPE dataset at test-time when trained with any other dataset. On FAUST, we are comparable with GeomFMap even though it is trained with ground truth correspondences.

We would like to stress that baselines such as 3D-CODED and GeomFMap require hundreds of SURREAL shapes, 2000 for 3D-CODED, in order to obtain reasonable results on SCAPE whereas we can obtain high quality results with significant improvement over GeomFMap with as low as 100 and 50 shapes. We stress that no other method is able to achieve such a generalization with this low number of shapes. We attribute our superior results over GeomFmap to a range of factors. First, in contrast to our unsupervised loss, GeomFmap uses a supervised loss without adequate regularization that leads to severe overfitting on challenging datasets with different poses such as SCAPE. This underscores the importance of enforcing structural properties of functional map in any loss function. Second, GeomFmap achieves a robustness to changes in shape orientation using *data augmentation* and ground truth functional map supervision, whereas we align the shapes manually. These experimental results confirm our findings that the devil in non-rigid shape matching lies in approximate *rigid* alignment and such weak supervision is equivalent to having supervised ground truth correspondence. Compared to GeomFmap, we obtain better results with ZoomOut as it refines initial maps better if they do not contain large errors, e.g. due to symmetries, which we observe with GeomFmap. Also, when the initial maps are good, the refined maps are often similar regardless of initial maps.

**Ablation Study** We show in Table 3 the ablation of our method trained on Surreal and tested on Faust and Scape. E3 (Laplacian commutativity) is the most important while E2 (Orthonormality) is the least among the three losses. Drastic decrease in performance of our method (All) without weak supervision underlines its importance. The drop is less severe in case of Faust where one axis is already aligned in contrast to Scape that is not aligned at all.

## 5.2 Deep Functional Maps with any Loss Function

The goal of this section is to unpack the minimum ingredients of a deep functional map pipeline such that it leads to unification of all the recent work under these minimum conditions. To this end, we test one representative deep functional map each from supervised setting and unsupervised setting with different loss functions. We optimize their loss functions with our PointNet++ feature extractor with low eigen basis (30) and our regularizers in both functional map pipeline and discard any other regularizer or feature extractor as proposed in these works. We train on SURREAL dataset from which we sample 500 shapes for training and test the trained models on test sets of FAUST re-meshed, SCAPE re-meshed.

**Unsup FMNet loss + Ours** Halimi et al. [2019] is an unsupervised approach that uses a soft correspondence based loss with geodesic matrix. Note that in their paper, Unsup FMNet relies on the SHOT descriptor that we replace with a PointNet++ feature extractor. We use their unsupervised loss in addition to our regularization terms.

**GeomFmap loss + Ours** We also evaluate Donati et al. [2020], a supervised approach where the ground truth functional map is computed in the spectral domain. We use this supervised loss function but discard their regularization proposed to alleviate overfitting. We also discard their feature extractor and do not perform any data augmentation. We sample 6000 vertices randomly for each shape as more vertices should lead to a better ground truth functional map estimation. *GeomFmap* simply reports the performance of Donati et al. [2020] without any modifications.

**Results and Discussion** We summarize the findings in Table 4. Remarkably, we obtain state of the art results with both loss functions. In particular, GeomFmap supervised spectral loss when optimized with our framework leads to significant increase in accuracy on the challenging SCAPE dataset. This shows the generalization capability of our framework. Similar performance boost is observed with Unsup FMNet on both datasets. It must be noted that memory footprint/training time of Halimi et al. [2019] is 50 times more as it requires either geodesic matrices to fit to RAM or load them on the fly for each pair. Our approach does not require Geodesic matrices, as in FMnet and UnSupFmnet, ground truth maps, as in GeomFmap and FMnet, regularizers, such as descriptor preservation in SurfmNet

Table 4: Comparative results of different loss functions when trained with our framework on Surreal and tested on remeshed Faust and Scape.

| Method \ Dataset | F | S |
|---|---|---|
| GeomFap+zo | 2.5 | 9.2 |
| Unsup FMNet loss + Ours | 6.3 | 7.7 |
| Unsup FMNet loss + Ours +zo | 4.4 | 5.2 |
| GeomFap loss + Ours | 5.0 | 7.7 |
| GeomFap loss + Ours +zo | 2.7 | 4.6 |
| Ours | 5.0 | 8.3 |
| Ours +zo | 2.8 | 5.5 |

Table 5: Comparative results on partial Shrec benchmark

| Method \ Dataset | Holes | Cuts |
|---|---|---|
| Litany et. al | 16 | **12** |
| Ours | **12** | 15 |

and regularized FMap layer in GeomFMap. Furthermore, when we remove these components from the respective works and include our minimum components, we get comparable or better results, thus proving the redundancy empirically.

## 5.3 Partial Shape Matching

Finally, we quantitatively evaluate our method in the partial matching scenario on the challenging SHREC'16 Partial Correspondence benchmark Cosmo et al. [2016]. The dataset is composed of 200 partial shapes (from a few hundred to 9K vertices each) belonging to 8 different classes (humans and animals), undergoing nearly-isometric deformations in addition to having missing parts of various forms and sizes. Each class comes with a "null" shape in a standard pose which is used as the full template to which partial shapes are to be matched. The dataset is split into two subsets, namely cuts (removal of a few large parts) and holes (removal of many small parts).

**Experimental Setup** The dataset contains several shapes whose number of points range from few hundreds to 2500. We use some of these shapes as a validation set and separate them from training or test set. Holes dataset is shown to be more challenging than cuts in Litany et al. [2017b]. Our loss function for partial shape matching does not contain any hyperparameters. Thus, we use validation set to only validate the training iterations. We consider Litany et al. [2017b] as our main baseline as it is considered state of the art for partial shape matching. Remark that no existing functional maps learning-based approach has yet been proposed for *partial* non-rigid shape matching.

**Results and Discussion** We present our findings on partial shape matching in Table 5. We obtain superior results on holes dataset. However, on cuts dataset, Litany et al. obtains better results. We attribute it mainly to the fact that convergence is found to be different for different shapes with our learned model. Thus, a shape in test set obtains optimal matching at time that is different for other shapes in test set. This could be due to a large fluctuations in the number of points per shape. Results shown here are obtained when our model was trained with a fixed number of iterations for whole test set. Note that the method of Litany et al. [2017b] is not learning based but relies on expensive manifold optimization for every pair of shapes at test time. In contrast, our method obtains a correspondence directly with pre-trained features and without the need for any test time optimization.

## 6 Conclusion and Future Work

We presented a novel weakly supervised method based on the functional map representation for both full and partial shape matching. Our main observation is that weak supervision in the form of approximately rigidly aligned input data is sufficient for learning powerful features to solve the non-rigid correspondence problem from raw data. Moreover, we establish that the key to cross dataset generalization lies in working with low number of eigen basis and enforcing very basic structural properties of a functional map. Our method for partial shape matching is also the first approach towards learning partial functional map and is of independent interest. We believe that this method will set the future direction of research, especially towards simpler techniques and weak supervision, in both near isometric as well as partial shape matching.

**Acknowledgement** Parts of this work were supported by the ERC Starting Grant StG-2017-758800 (EXPROTEA), ANR AI Chair AIGRETTE, and a gift from Nvidia. We thank anonymous reviewers for their helpful comments and Ruqi Huang and Simone Melzi for their help in performing quantitative comparisons on the Partial SHREC dataset.

## 7 Appendix

We have shown empirically in the ablation study in Table 3 that each regularizer contributes to the overall performance and is thus necessary. Note that orthogonality and commutativity does not imply bijectivity. One counter example is Identity and $-$ Identity matrix. Both are orthogonal and would commute with any Laplacian. However, they are not inverse of each other.

In Figure 1 and 2, we show the corresponding curves below that are consistent with average geodesic error shown in Section 5. Figure 3 and 4 shows some examples of rigid alignment that we refer to as weak supervision in our paper

Figure 1: Error on Scape when trained on Surreal   Figure 2: Error on Faust when trained on Surreal

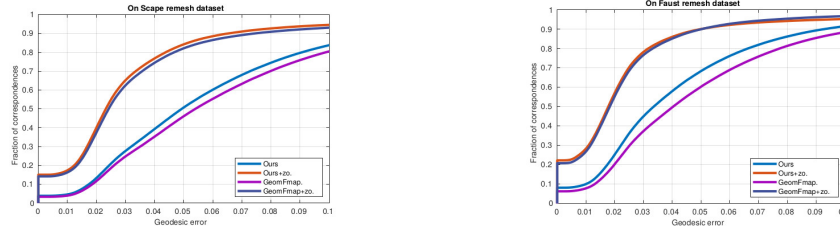

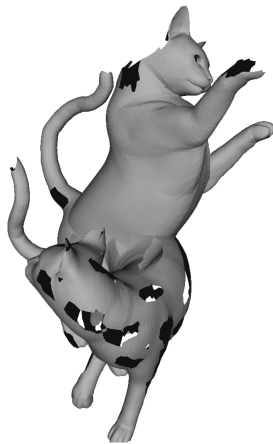

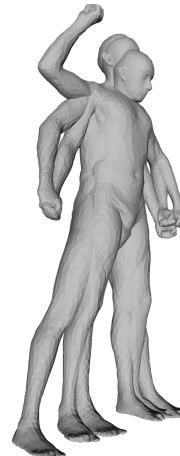

Figure 3: An example from Partial SHREC   Figure 4: An example of alignment from SCAPE

## 8 Broader Impact

Efficient algorithms for solving the shape correspondence problem have immediate impact in many areas of science and engineering from statistical shape analysis, creation of virtual avatars, medical imaging (for instance for detecting anomalies, and performing follow-up analysis). Our approach can immediately be adapted and tested in such diverse scenarios. This is particularly true as our method is efficient and fully automatic. We believe that the observations made in our work can also lead to new insights in other areas including graph matching and machine translation, where data is often represented as point clouds in some embedding space.

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
