[Reviews · NeurIPS 2020]

Review 1

Summary and Contributions: The paper introduces a weakly supervised framework for shape matching relying on functional maps. The framework includes learning of feature descriptors directly from raw data using a weak supervision (shapes just need to be approximately rigidly aligned), then uses an unsupervised loss for full shape matching. It tackles the partial shape matching too via basis alignment, and in an experimental way it is shown to provide state of the art results with different loss functions.

Strengths: The paper provides state-of-the-art results using a weakly supervised approach, outperforming even many of the fully supervised approaches. It is theoretically well grounded and its evaluation includes all recent state-of-the-art methods to the extent of my knowledge. The shape matching problem finds many applications in e.g. robotics and VR, thus I consider it interesting and relevant fort the NeurIPS community.

Weaknesses: A couple of details which I found not strictly limiting but not 100% convincing (i.e. they should be better motivated IMO): - in Table 1, the presented method outperforms GeomFmap only when applied together with ZoomOut. The difference becomes even bigger when it is tested against SCAPE vs FAUST (e.g. with FAUST the difference in performances between the two methods is not huge, and when ZO is applied it becomes zero; with SCAPE the difference is bigger, and when ZO is applied the presented method performs better than GeomFmap). How much of the improvement is due to the newly introduced method, how much to ZO, and how much is related to the dataset it is tested on? - the presented method outperforms Litany et al’s on partial SHREC benchmark holes. It does not on cuts. I think it would have been more transparent to provide the negative result in the paper with a discussion on why that happened (discussion which I read in the supplementary material and looks reasonable to me) rather than showing only the positive one in the paper.

Correctness: Claims and methodology look correct to me. I would like to see the negative partial matching result in the table for transparency. (In my opinion this does not limit the positive contributions of the paper)

Clarity: The paper is well written overall, there are just few typos that should be corrected (e.g. faircomparison in line 212, of k1, k2 of in line 110, etc)

Relation to Prior Work: Yes

Reproducibility: Yes

Additional Feedback: After reading both the rebuttal and discussing with reviewers, I decided to lower my score to 7 (accept). I am still overall positive about the paper but understood that there are few points that should be made more clear.


Review 2

Summary and Contributions: This paper presents a weakly supervised approach for shape matching. The weak supervision comes from rigidly (manually) aligning the shapes into a canonical orientation. The paper builds heavily on previous work (Donati etal, Ovsjanikov etal). The paper claims to provide minimum and sufficient conditions in terms of loss functions, learnt descriptor and pre-processing (rigid alignment) that can achieve better performance than baselines. Experiments are done on dataset used in shape matching tasks.

Strengths: There are three main claims in the paper. First, weak supervision in the form of manually aligning shapes to a canonical orientation helps in shape matching. Second, using low dimensional basis function and enforcing structure properties of functional maps helps in generalization across datasets. Thirdly, a loss function is proposed for partial shape matching. The importance of all these claims are well described in the paper.

Weaknesses: Some of the claims made in this paper have already been explored in previous works or are rather obvious. 1. PointNet features are not rotation invariant. We can try to make them robust by using data-augmentation. Rigidly aligning the shapes will obviously make the features consistent across shapes, especially when shapes share same human template. Also, note that Donati etal used learnable point based features, so it is not a new contribution, as is claimed in Line-44. 2. The importance of functional map representation with reduced basis has already being shown in Donati etal (See Section 2). 3. The importance of structural properties of the functional maps are well justified in the paper. Though, the loss functions have already been used in previous works and their efficacy has been described in previous works. 4. The paper also claims an independent contribution on partial shape matching, though effective, seems out of the place in the paper. It would have been more interesting paper, if it focused on one aspect. Even focusing only on partial shape matching and detailed analysis on only one aspect would have been a good contribution to the community.

Correctness: 1. There are a few ablation study that should be done to verify the validity of this method. For example, for the unsupervised losses, how much each of them contribute to the final performance? Also, how much gain in the performance in the Table 2 is because of weak alignment and how much is because of weak alignment + unsupervised losses. 2. What is the performance of Donati etal [2020] when shapes are aligned under weak supervision? This will validate the importance of weak supervision. Perhaps, it is already been taken care of in Table 3, but it is not very clear. 3. The performance of GeomFmap without zoom-out is better its counterpart in proposed method. Why is that?

Clarity: Paper is well written. There is a minor grammatical error on Line 24.

Relation to Prior Work: Prior works are well covered and relevance is well explained throughout the paper. I appreciate this about the paper.

Reproducibility: Yes

Additional Feedback:


Review 3

Summary and Contributions: This paper proposes a deep architecture that learns to regress the functional map given two point clouds. It leverages learned descriptors, a set of regularizers on the functional map and an unsupervised loss both for the full-to-full and partial-to-full setting. The paper finds that the alignment of a major axis is fundemantal to get high accuracies and reports state of the art results.

Strengths: This paper nicely combines a lot of state-of-the-art findings & techniques in functional maps within an end-to-end learning framework to produce improved accuracy while removing the requirement of supervising the functional correspondence. I also like the idea of learning descriptors along with the functional correspondence. The experimental observation that 'rigid alignment' is the key in getting high accurate functional correspondence further motivates methods that can simultaneously solve for both.

Weaknesses: - Abstract is unclear and does not hint about the contributions of the paper. 'Sufficient' and 'minimum' are technically involved adjectives and I would refrain from using them without proofs of conditions that they refer to. - I am not sure whether rigid alignment of deformed bodies is a trivial problem to solve. In this sense, it might be hard to use it as a ‘weak supervision’. While the current datasets are acquired under such control, it is not obvious how the real applications would establish that. - Can’t we enforce orthonormality as restricting C to be on the Stiefel manifold? Why do we need this as a regularization? - Furthermore, the paper does not only constrain individual Cs to be orthonormal+commutative but constrains the composition of the two: C_{12}^T C_{21}. This amounts to introducing bijectivity into the orthonormalization+Laplacian regularizers, probably making the former regularizer (bijectivity) redundant. Can we have a discussion on this? - While the introduction heavily stresses that ‘weak supervision’ by ‘rigid alignment’ is a requirement, the method and technical discussions contain no hints as to why this is the case. The use of this weak supervision signal is also not explained in the method section. From this, I understand that this was only an experimental finding. As such, it cannot be a central contribution in this paper - implicitly, many existing works use this supervision because they work on the same datasets. I think this is a general issue in the method section where the contributions are not correctly prioritized. - Is it possible to have theoretical argumentation as to why rigid alignment is necessary? - Similarly, why (from a theoretic perspective) using a small amount of eigenbases aids generalization? In general, the technical contributions of the paper pose little novelty while the results seem to be on par or better with supervised approaches. However, I notice that when prior work is used with the proposed losses the results are also on par or better. This makes me think that for the given data, this problem is rather easy to solve in an unsupervised manner. Hence, the improvement may not stem from the combination of enlisted technical points. Finally, how about GeomFmap with a deep descriptor + the unsup. loss + regularizers ? Could the authors also comment on solving the partial-to-partial mapping?

Correctness: From a technical perspective, the proposed method is heavily based on prior work. The evaluations are on well established datasets and uses accepted metrics in the literature. As such, I believe the paper is technically and empirically correct.

Clarity: The paper contains some grammatical errors but in general it is readable and easy to follow.

Relation to Prior Work: While the major related works are covered, the paper misses out some relevant works that seem to be in parallel direction. For instance, Zhangsihao et al. alleviates the need for SHOT descriptors: Yang, Zhangsihao, Or Litany, Tolga Birdal, Srinath Sridhar, and Leonidas Guibas. "Continuous Geodesic Convolutions for Learning on 3D Shapes." arXiv preprint arXiv:2002.02506 (2020).

Reproducibility: Yes

Additional Feedback: * ‘It is still not clear what are minimum and sufficient ingredients of a deep functional map pipeline…’ -> ‘It is still not clear what the minimum and sufficient ingredients of a deep functional map pipeline are…’ . This also happens in introduction. Such error is observed also in other places within the text. * ln. 24: ‘been recently been’ -> ‘been recently’ * Ln 30. ‘pipeline.’ -> ‘pipeline?’, ‘maps.’ -> ‘maps?’ * off(.) is not defined
 As the authors note in the rebuttal, they have addressed the points of R2 and R4. However, I must admit that I remain unconvinced. I would like to rephrase and remind the issues that I raised: - The bijectivity does not follow from orthonormality, but the authors are not just imposing orthonormality on a single matrix. They do it on the composition of the two. There is no proof of the independence of these regularizers. Again, I could be wrong that there are simpler functions implementing those regularizers, but I would just like an argumentation on this front. - Introduction and abstract has to be tuned down as the paper has no theoretical point on the sufficient and minimum (probably meaning necessary) conditions. This might be done in a camera ready, at the expense of diminishing the contributions of the paper. - On the rigid alignment: There can be more than meets the eye here. A vector field in general is a superimposition of rigid and non-rigid deformations. Cancelling out one term can make room for other regularizations on the map such as pure non-rigidness. I like the idea, but I don’t think that the method explores it well enough.


Review 4

Summary and Contributions: The paper studies deep functional map for shape matching. It proposes an end-to-end framework to directly learn feature descriptors from raw data with rigid shape alignment as a form of weak supervision. It shows a minimum and sufficient condition for learning good maps in the deep functional map framework and demonstrates good matching results on several popular shape matching datasets. The paper also studies how to conduct partial shape matching which requires a different regularization compared with the full shape matching problem.

Strengths: One contribution of the paper is to demonstrate that good mappings can be obtained without the need of ground truth maps as long as proper regularizations are incorporated. It studies the difference of full shape matching and partial shape matching and reflect the differences in their different loss design. Although the theoretical study has been done previously and the regularizations this paper adopts are not new at all, demonstrating how to incorporate them into an end-to-end trainable network to achieve good matching results is still valuable. The paper is clearly written and is technically sound, though it might be a bit heavy to readers not familiar to functional maps. The experimental results partially support the claimed contribution.

Weaknesses: The paper adopts a lot of existing techniques such as point feature extraction and functional map regularization in a weakly supervised deep learning scenario, which as a whole is a contribution but quite incremental. Also the experimental results are not very solid. When people are evaluating shape matching results, usually different metrics are provided since it is hard to use a single number to say one matching algorithm is better. Among them, the percentage of correct correspondences v.s. geodesic error curve is the most popular one since it shows a full spectrum of the matching quality, which is missing in the experiments. It is not very convincing to say the proposed method is better or on par with previous approaches just from the average error alone. Also I do not see why the ingredients proposed in this paper can be treated as the minimum ingredients of a deep functional map pipeline. Is their any theoretical proof for this? Otherwise the claim should be down tuned.

Correctness: The method is correct while the “minimum and sufficient ingredients” claim is not very convincing to me.

Clarity: Yes

Relation to Prior Work: Yes

Reproducibility: Yes

Additional Feedback: There are several ablation studies would strengthen the paper, for example showing the connection between the mapping quality and how partial the shapes are in a deep network, studying the robustness the resulting maps to data corruption. Although I am not fully convinced by some of the claims and experimental results, I still value the end-to-end weakly supervised deep functional map for both full shape and partial shape matching as a whole. ========After Rebuttal======== The authors have addressed some of my concerns. After discussing with the other reviewers, I decided to keep my score unchanged and I expect the authors to incorporate the clarifications in their final version and also tune down the claims in intro and abstract regarding the sufficient and minimum conditions.

[Author Response · NeurIPS 2020]

We thank all the reviewers for their comments, and acknowledging that our approach is R1: "theoretically well grounded,
state-of-the-art using a weakly supervised formulation". R4: "shows that good mappings can be obtained as long as
proper regularizations are incorporated which is valuable. End-to-end weakly supervised deep functional map for both
full shape and partial shape matching as a whole is valuable". In the following, we address major concerns:
**Non-linear Effect of Zoomout on accuracy (R1)** In Table 1 (main paper), compared to GeomFmap, we obtain better
results with zoomout as it refines initial maps better if they do not contain large errors, e.g. due to symmetries (detailed
next), which we observe with GeomFmap when trained on Faust and tested on Scape (non-aligned dataset). Also, when
the initial maps are good (in range 3-5), refined map is often similar regardless of initial maps.
**Weak Supervision (R2,R3)** It is necessary due to the presence of symmetries. Since some poses (e.g. the neutral pose)
are fully extrinsically symmetric, a PointNet like feature extractor cannot distinguish left/right unless the shapes are
aligned, we need *some way* to disambiguate them for correspondence. Therefore some amount of weak supervision, such
as rigid alignment, is necessary and also explains performance drop of GeomFmap when tested on Scape(non-aligned)
**Ablation, R1,R2**: We show below ablation of our method trained on Surreal and tested on Faust and Scape. E3

| Losses | All | E1 | E2 | E3 | (E1+E3) | All-not-aligned |
|---|---|---|---|---|---|---|
| **Scape** | 7.5 | 12 | 15 | 9.5 | 8.2 | 20 |
| **Faust** | 5.2 | 11 | 14 | 9.0 | 6.3 | 8.0 |

Table 1: Avg. Geodesic error with individual losses with and without alignment when trained with Surreal.
(Laplacian commutativity) is the most important while E2 (Orthonormality) is the least among the three losses. Drastic
decrease in performance of our method (All) without weak supervision underlines its importance. The drop is less
severe in case of Faust where one axis is already aligned in contrast to Scape that is not aligned at all. Table 3 in the
paper also validates the effectiveness of weak supervision on GeomFmap and Unsup FMNet as they are trained on
aligned dataset. We will clarify this more in paper.
**Sufficient and Minimum Conditions (R3,R4)** Our approach does not require Geodesic matrices, as in FMnet and
UnSupFmnet, ground truth maps, as in GeomFmap and FMnet, regularizers, such as descriptor preservation in SurfmNet
and regularized FMap layer in GeomFMap. Removal of so many components without compromising results motivated
us to use this terminology. Furthermore, when we remove these components from the respective works and include
our minimum components, we get comparable results, thus proving the redundancy. We always claimed this based on
empirical findings. Even in the abstract, we mention "with slight of abuse of notation." We will tone it down further to
avoid any possibility of this being a theoretical condition and pose the in-depth theoretical study as a future work.
**Weakness by R2**: We believe there is a misunderstanding. In line 44, we claim to achieve state-of-the-art results from
point clouds with rigid supervision when compared to Donati et al. that learns it with full *point to point ground truth*
*dense correspondences*. We never claim our approach to be the first one to learn from point clouds. We do not agree
with the assessment that contribution on partial shape matching seems out of place in the paper. We believe end-to-end
learning pipeline that can handle both partial and full shape matching is a valuable contribution to the community.
**Weakness by R3**: We need orthonormal $C$ as it promotes locally area preserving correspondences. One can enforce
the same with Stiefel manifold but the resulting problem is much harder to optimize and besides, as shown in our
partial matching results, does not bring additional accuracy. Note that bijectivity does *not* follow from enforcing
commutativity and orthonormality. We will be happy to provide analytical counter examples to prove this. Low no. of
Laplacian eigen-basis reduces overfitting and helps generalization as the embedding space decreases and bias/variance
trade off kicks in. These may be simple observations but have a significant impact, as our approach with as low as
100 approximately rigidly aligned shapes obtains comparable results to much more expensive methods that require
thousands of densely annotated maps. We do not understand what does 'GeomFmap + unsup. loss + regularizers'
means? Our Unsup. loss only consists of regularizers. Besides, Table 3 contains GeomFmap with a deep descriptor and
our unsupervised loss on aligned dataset with name 'GeomFmap loss+Ours'.
**Cuts result/Discussion in supplement, R1** We agree and will include them in the main text. Thank you for suggestion.
**Comparison with a different metric, R4**: We show the corresponding curves below that are consistent with avg. error.
We thank all reviewers for pointing out typos and will fix them. We addressed all major concerns of R2 and R4 and
thus, kindly ask them to **reconsider their ratings based on rebuttal**.



[Meta-Review · NeurIPS 2020]

This paper proposes a deep learning approach to predicting functional maps for shape matching. The reviewers agree that the paper makes a useful contribution, is clearly presented, and should be published. They raise concerns in their reviews and in the discussion about two issues that we strongly recommend the authors address in the camera ready version. First, the use of technical terms like "minimum" and "sufficient" imply theoretical claims that are unproved. Second, the derivation of the method needs to be more carefully presented. For example, the paper does not consider that the constraints imposed amount to imposing bijectivity on the composition of two matrices. See R3's review for more details.